

# miRDRN—miRNA disease regulatory network: a tool for exploring disease and tissue-specific microRNA regulatory networks

Hsueh-Chuan Liu[1], Yi-Shian Peng[1] and Hoong-Chien Lee[1,2]

[1] Department of Biomedical Sciences and Engineering, National Central University, Taoyuan City, Taiwan
[2] Department of Physics, Chung Yuan Christian University, Zhongli District, Taoyuan City, Taiwan

Corresponding author
Hoong-Chien Lee,
hclee12345@gmail.com

## ABSTRACT

**Background:** MicroRNA (miRNA) regulates cellular processes by acting on specific target genes, and cellular processes proceed through multiple interactions often organized into pathways among genes and gene products. Hundreds of miRNAs and their target genes have been identified, as are many miRNA-disease associations. These, together with huge amounts of data on gene annotation, biological pathways, and protein–protein interactions are available in public databases. Here, using such data we built a database and web service platform, miRNA disease regulatory network (miRDRN), for users to construct disease and tissue-specific miRNA-protein regulatory networks, with which they may explore disease related molecular and pathway associations, or find new ones, and possibly discover new modes of drug action.

**Methods:** Data on disease-miRNA association, miRNA-target association and validation, gene-tissue association, gene-tumor association, biological pathways, human protein interaction, gene ID, gene ontology, gene annotation, and product were collected from publicly available databases and integrated. A large set of miRNA target-specific regulatory sub-pathways (RSPs) having the form $(T, G_1, G_2)$ was built from the integrated data and stored, where $T$ is a miRNA-associated target gene, $G_1$ ($G_2$) is a gene/protein interacting with $T$ ($G_1$). Each sequence $(T, G_1, G_2)$ was assigned a $p$-value weighted by the participation of the three genes in molecular interactions and reaction pathways.

**Results:** A web service platform, miRDRN (http://mirdrn.ncu.edu.tw/mirdrn/), was built. The database part of miRDRN currently stores 6,973,875 $p$-valued RSPs associated with 116 diseases in 78 tissue types built from 207 diseases-associated miRNA regulating 389 genes. miRDRN also provides facilities for the user to construct disease and tissue-specific miRNA regulatory networks from RSPs it stores, and to download and/or visualize parts or all of the product. User may use miRDRN to explore a single disease, or a disease-pair to gain insights on comorbidity. As demonstrations, miRDRN was applied: to explore the single disease colorectal cancer (CRC), in which 26 novel potential CRC target genes were identified; to study the comorbidity of the disease-pair Alzheimer's disease-Type 2 diabetes, in which 18 novel potential comorbid genes were identified; and, to explore possible causes

that may shed light on recent failures of late-phase trials of anti-AD, *BACE1* inhibitor drugs, in which genes downstream to *BACE1* whose suppression may affect signal transduction were identified.

## INTRODUCTION

Protein–protein interactions (PPIs) are critical to almost all biological process, and a good knowledge of the network of interacting proteins is crucial to understanding cellular mechanisms (*Rual et al., 2005*). Recent advances in biotechnology, such as high-throughput yeast two-hybrid screening, have allowed scientists to build maps of proteome-wide PPI, or interactome. Conventionally, a PPI map is a static network, in which each node represents a protein and an edge connecting two proteins indicates that there is experimental evidence showing that, under certain circumstances, the two proteins would interact. In reality, a PPI network (PPIN) should be viewed as a dynamic entity: it is an interaction network that is intrinsically controlled by regulatory mechanisms and changes with time and space (*Liang & Li, 2007*), as determined by the physiological condition of the cell in which the proteins reside. If there is a PPIN that includes all possible PPIs, then, under a specific physiological condition only a specific sub-network of the PPIN is realized.

MicroRNAs (miRNAs) are small (~22 nucleotides) noncoding regulatory RNA molecules in plants, animals, and some viruses. In a process known as RNA interference, a miRNA regulates gene expression by destabilizing and/or disrupting the translation of fully or partially sequenced mRNA (*Bartel, 2009*; *Landgraf et al., 2007*). In this way a miRNA regulates the formation of all PPINs to which its target is connected, and by extension all biological processes (BP) with which those PPINs are involved. As well as acting as a tumor suppressor gene (TSG), a miRNA may also act as an oncogene, say, by targeting a TSG (*Zhang, Dahlberg & Tam, 2007*). The function of a specific biological process, or its malfunction, such as associated with a disease, typically involves a complex composed of a set of miRNA-regulated proteins, together with their interacting protein partners. The study of such miRNA-protein complexes should be an integral part of understanding BP (*Hsu, Juan & Huang, 2008*) as well as diseases.

An understanding of the molecular and physio-pathological mechanisms of diseases is crucial for the design of disease preventive and therapeutic strategies. The combination of experimental and computational methods has led to the discovery of disease-related genes (*Botstein & Risch 2003*; *Kann, 2010*). An example is the causal relation connecting the malfunction causing mutations in the enzyme phenylalanine hydroxylase to the metabolic disorder Phenylketonuria (*Scriver & Waters, 1999*). Many human diseases

cannot be attributed to single-gene malfunctions but arise from complex interactions among multiple genetic variants (*Hirschhorn & Daly, 2005*). How a disease is caused and how it can be treated can be better studied on the basis of a body of knowledge including all associated genes and biological pathways involving those genes.

Diseases are usually defined by a set of phenotypes that are associated with various pathological processes and their mutual interactions. Some relations between phenotypes of different diseases may be understood on the basis of common underlying molecular processes (*Barabási, Gulbahce & Loscalzo, 2011*), such as when there are genes associated with both diseases. It has been shown that genes associated with the same disorder encode proteins that have a strong tendency to interact with each other (*Goh et al., 2007*). More specifically, one may consider two diseases to be related if their metabolic reactions within a cell share common enzymes (*Lee et al., 2008*). Networks of PPIs have also been studied in the context of disease interactions (*Ideker & Sharan 2008*; *Lim et al., 2006*).

Here, we report on a web service platform, miRNA disease regulatory network (miRDRN) (http://mirdrn.ncu.edu.tw/mirdrn/). The platform contains two parts, a database that stores a set of newly constructed set of 6,973,875 $p$-valued target-specific regulatory sub-pathways (RSPs) associated with 116 diseases in 78 tissue types built from 207 diseases-associated miRNA regulating 389 genes; and a novel web-based tool that, using the RSPs stored in miRDRN and information from miRNA-related databases, facilitates the construction and visualization of disease and tissue-specific miRNA-protein regulatory networks for user specified single diseases and, for comorbidity studies, disease-pairs. We demonstrate three applications of miRDRN: to explore the molecular and network properties of the single disease colorectal neoplasm; to study the comorbidity of the disease-pair Alzheimer's disease-Type 2 diabetes (AD-T2D); and, by using miRDRN to construct a miRNA regulatory sub-network centered on the gene *BACE1*, to look for insights that may explain why several anti-AD, *BACE1* inhibiting drugs that failed recent late-phase trials worsened conditions of treatment groups. We believe findings from miRDRN, even exploratory in nature, may potentially lead to the identification of new drug targets and new understanding in modes of drug action.

## MATERIALS AND METHODS

### Data integration

miRDRM integrated data from several existing database on disease-miRNA association, miRNA-target gene association, gene ontology, biological pathway, and PPI. For disease-specific cases disease-associated miRNAs and targets were obtained from human microRNA and disease associations database (HMDD) (*Li et al., 2014*) (v2.0, http://www.cuilab.cn/hmdd). For non-disease specific cases, miRNA/siRNA and targets were obtained from TarBase (*Vlachos et al., 2015*) (v7.0, http://carolina.imis.athena-innovation.gr/diana_tools/web/index.php?r=tarbasev8%2Findex/). In disease-specific cases, the optional filter requiring miRNA-target association be assay validated used TarBase data; the filter excludes miRNA-target pairs appearing in HMDD but not in TarBase (if and when this happens). Gene-tissue associations were taken from NCBI-Entrez (*NCBI Resource Coordinators et al., 2018*); RSP associations with known pathways from the

**Table 1 Databases used in the construction of miRDRN and data usage.**

| Database | Information used | Where used | Reference |
|---|---|---|---|
| HMDD 2.0 | Disease-miRNA association, miRNA-target association | Query interface (disease selection) | *Li et al. (2014)* |
| TarBase 7.0 | miRNA-target association miRNA-target validation | Query interface (miRNA/siRNA selection) Query interface (filter—miRNA-target validation) | *Vlachos et al. (2015)* |
| NCBI-Entrez | Gene-tissue association | Query interface (filter—tissue) | *NCBI Resource Coordinators et al. (2018)* |
| KEGG | Biological pathways | Query interface (filter—pathway) | *Kanehisa et al. (2017)* |
| TAG | Gene-tumor association | Query interface (filter—tumor genes) | *Chen et al. (2013)* |
| NCBI-GeneBank | Gene ID, TF, and/or RC | Query interface (filter—TF and/or RC) | *NCBI Resource Coordinators et al. (2018)* |
| NCBI-OMIM | OMIM ID | Target genes; genes on regulatory network (activated by mouse-click) | *NCBI Resource Coordinators et al. (2018)* |
| BioGRID | Protein–protein interaction | Construction of regulatory sub-pathways | *Chatr-aryamontri et al. (2013)* |
| GO | Biological processes, molecular functions; gene ontology, gene annotation, and product | Computation of Jaccard index for regulatory sub-pathways; annotation of genes on regulatory network (activated by mouse-click) | *Ashburner et al. (2000)* |

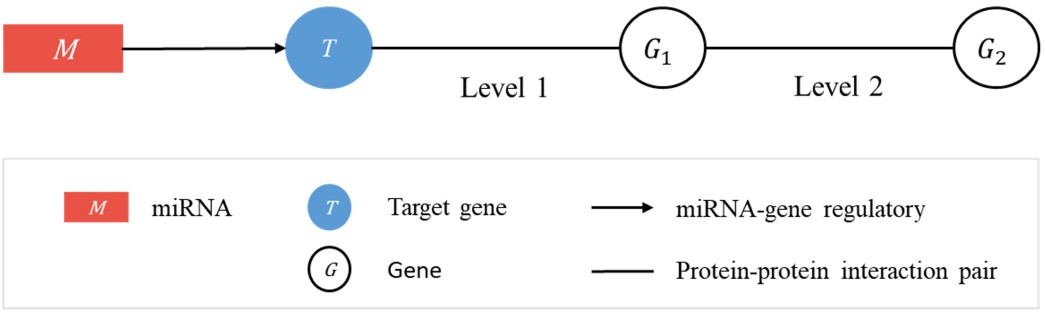

**Figure 1 Regulatory sub-pathways.** In the linked sequence $(M, T, G_1, G_2)$, called a miRNA-specific regulatory sub-pathway (MRSP), $M$ is a miRNA, $T$ is its regulatory target gene, $G_1$ is a protein interacting (according to PPI data) with $T$, and $G_2$ is a protein interacting with $G_1$. In the text the sequence $(T, G_1, G_2)$ is called a target-specific regulatory sub-pathway, or simply, regulatory sub-pathway (RSP).

Kyoto Encyclopedia of Genes and Genomes (KEGG) (*Kanehisa et al., 2017*) (http://www.genome.jp/kegg); gene-tumor associations from TAG (*Chen et al., 2013*) (http://www.binfo.ncku.edu.tw/TAG); OMIM IDs of target genes and genes on regulatory networks from NCBI-OMIM (*NCBI Resource Coordinators et al., 2018*) (https://www.ncbi.nlm.nih.gov/omim); gene IDs and transcription factor and/or receptor from NCBI-GeneBank (*NCBI Resource Coordinators et al., 2018*) (https://www.ncbi.nlm.nih.gov/genbank); data on human PPI from biological general repository for interaction datasets (BioGRID) (*Chatr-aryamontri et al., 2013*) (https://thebiogrid.org/); and gene ontology, gene annotation and gene product information from GO (*Ashburner et al., 2000*) (http://geneontology.org/). (Table 1).

## Construction of miRNA-associated target-specific regulatory sub-pathways

Consider a linked sequence $(M, T, G_1, G_2)$ (Fig. 1), where $M$ is a miRNA, $T$ is its regulatory target gene, $G_1$ is a gene whose encoded protein ($p_1$) interacts (according to PPI data) with

the protein ($p_T$) encoded by $T$, and $G_2$ is a gene whose encoded protein ($p_2$) interacts with $p_1$. In what follows, when there is little risk of misunderstanding, the same symbol will be used to represent a gene or the protein it encodes. We call the sequence ($T, G_1, G_2$) a target-specific RSP, or simply a RSP, and ($M, T, G_1, G_2$) a miRNA-specific RSP (MRSP). Given a target gene $T$, we use PPI data from BioGRID to collect all RSPs by extending from $T$ two levels of interaction.

## Jaccard score of a regulatory sub-pathway

Jaccard similarity coefficients (*Ng, Liu & Lee, 2009*) were used to score the RSPs, based on the assumption that there is a tendency for two directly interacting proteins to participate in the same set of BP or share the same set of molecular functions (MF). Given two sets $S1$ and $S2$ (in the current application, a set will be either a list of BP or a list of MF, both according to GO), the Jaccard (similarity) coefficient (JC) of $S1$ and $S2$ is defined as,

$$\mathrm{JC}(S1, S2) = \frac{|S1 \cap S2|}{|S1 \cup S2|}$$

Where $\cup$ is the union (of two sets), $\cap$ is the intersection, and $|Z|$ is the cardinality of $Z$. JC, which ranges from zero to one, is a quantitative measure of the similarity between two sets. For example, when $S1 = \{a, b, c\}$ and $S2 = \{b, c, d\}$, JC ($S1, S2$) = 2/4 = 0.5.

Let ($T, G_1, G_2$) be an RSP as defined in the previous section and denote by $[G]$ the set of BP (or pathways) (*Kanehisa et al., 2017*; *Ashburner et al., 2000*) that involve the gene $G$. We define the Jaccard score, or JS, of RSP as,

$$\mathrm{JS_X}(T, G_1, G_2) = \frac{1}{2}\left(\mathrm{JC}\big([T]_X, [G_1]_X\big) + \mathrm{JC}\big([G_1]_X, [G_2]_X\big)\right)$$

Where X may be BP or MF. If the pair $[T]$ and $[G_1]$ do not share a common term, then the corresponding JC has a zero value; similarly for the pair $[G_1]$ and $[G_2]$. In either case the RSP is considered to be not viable and discarded. In other words, miRDRN excludes any RSP with zero JC score.

## *p*-Value of a regulatory sub-pathway

A *p*-value for an RSP ($T, G_1, G_2$) was assigned as follows. Let the total number of BP (or MF, as the case may be) terms be $N$, and the number of terms in $[T]$, $[G_1]$, $[G_2]$, $[T] \notin [G_1]$, $[G_1] \notin [G_2]$ be $x$, $y$, $z$, $n_1$, and $n_2$, respectively, then the *p*-values, $P_1$ and $P_2$, for ($T, G_1$) and ($G_1, G_2$) are, respectively

$$P_1 = \frac{C_{n_1}^N C_{x-n_1}^{N-n_1} C_{y-n_1}^{N-x}}{C_x^N C_y^N}$$

and

$$P_2 = \frac{C_{n_2}^N C_{y-n_2}^{N-n_2} C_{z-n_2}^{N-y}}{C_y^N C_z^N}$$

The *p*-value for the RSP was set to be the greater of $P_1$ and $P_2$.

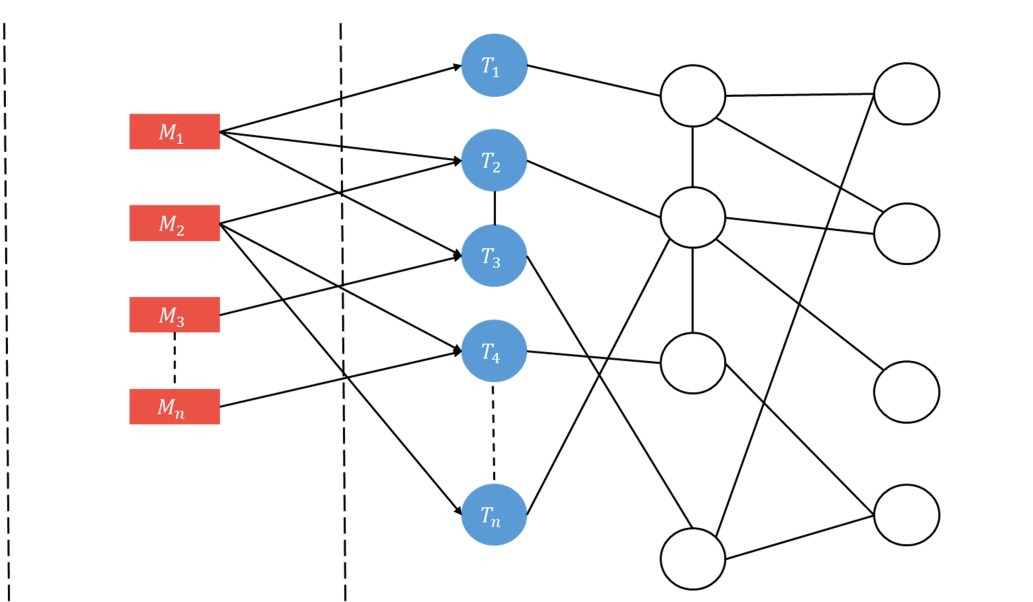

Figure 2 **Schematic construction of disease-specific miRNA regulatory network (RRN).** For a given disease there may be more than one miRNA associated with it (A), and each disease-associated miRNA may have one or more target genes (B). After all the miRNA-specific RSPs having the from $(M, T, G_1, G_2)$ are constructed (in the case of miRDRN utilization, retrieved from its database), an RRN is built from entire set of MRSPs by linking all unlinked pairs of genes/proteins if they have interaction according to BioGRID.

## Assembly and storage of target-specific regulatory sub-pathways

A union set of miRNA-associated target genes were collected from HMDD and TarBase and for every target a complete set of RSPs, with BP- and MF-type JC scores and $p$-values assigned, was assembled. The entire set of RSPs for all targets was stored in miRDRN.

## Construction of disease-specific miRNA regulatory network

A user-initiated construction of a disease-specific miRNA regulatory network (RRN) proceeds as follows. Step 1. Select a disease. Step 2. Collect from HMDD all miRNAs ($M$'s) and target genes associated with the disease. Step 3. For each $M$ and each of its targets retrieve from miRDRN storage all target-specific RSPs, thus forming a set of MRSPs. The union of the sets of MRSPs over all $M$'s is the set of disease-specific MRSPs for the selected disease. Step 4. Construct the disease-specific RRN from the set of disease-specific MRSPs by linking all unlinked pairs of genes/proteins if they have interaction according to BioGRID (Fig. 2).

## RESULTS

### miRNA disease regulatory network (miRDRN)—A database and web service platform

We built miRDRN (http://mirdrn.ncu.edu.tw/mirdrn/), a web-based service that allows the user to construct a disease and tissue-specific, $p$-valued, miRNA-protein

**Table 2 Data contained in current version miRDRN.**

| Type of data | Disease | miRNA | miRNA regulated gene | Target-specific RSP |
|---|---|---|---|---|
| Number | 116 | 207 | 389 | 6,973,875 |

**Table 3 Comparison of miRDRN with other miRNA-related databases.**

| Information | Database | | | | | |
|---|---|---|---|---|---|---|
| | miRDRN | HMDD *Vlachos et al. (2015)* | TarBase *Kanehisa et al. (2017)* | PhenomiR *Ruepp, Kowarsch & Theis (2012)* | miRwayDB *Das, Saha & Chakravorty (2018)* | miRPathDB *Backes et al. (2017)* |
| Disease-associated miRNA | Yes | Yes | – | Yes | Yes | – |
| miRNA-associated target gene | Yes | Yes | Yes | – | Yes | – |
| Target gene-KEGG association | Yes | – | – | – | Yes | – |
| Pathway-associated miRNA | Yes | – | – | – | – | Yes |
| Target-specific regulatory sub-pathway (RSP) | Yes | – | – | – | – | – |
| Disease-specific miRNA-protein regulatory network | Yes | – | – | – | – | – |
| Gene annotation (GeneBank, TAG, KEGG, GO) on all genes on network | Yes | – | – | – | – | – |
| Comorbidity of disease-pair | Yes | – | – | – | – | – |

regulatory network, or RRN. The current version of miRDRN contains 6,973,875 *p*-valued target-specific RSPs constructed through 389 miRNA-regulated genes from 207 diseases-associated miRNAs associated with 116 diseases (Table 2).

## Comparison of miRDRN with other miRNA-related databases

A number of databases and/or web service platforms on miRNA-related topics are publicly available (Table 3). Aside from HMDD and TarBase on which miRDRN was built (Table 1), PhenomiR (*Ruepp, Kowarsch & Theis, 2012*) is a database on disease-miRNA association, miRwayDB (*Das, Saha & Chakravorty, 2018*) is a database on disease-miRNA-target and target-KEGG term association, and miRPathDB (*Backes et al., 2017*) is a database on miRNA-pathway association. New and unique as a database, miRDRN stores the 6,973,875 *p*-valued target-specific RSPs it has assembled (Table 2). As a web service platform miRDRN is a tool that facilitates the construction and visualization of disease-specific RRNs using these RSPs in combination with resources from HMDD, TarBase, and several other databases (Table 1).

## Brief description of usage of miRDRN

miRNA disease regulatory network is reasonably user friendly; its many features are easily discovered by user exploration. Here, we give a brief description of its main features.

User may use miRDRN to explore a single disease, or the comorbidity of a disease-pair. In the course of either type of study, all relevant miRNAs, genes, and RSPs are made accessible to the user in tabulated form, and RRNs in the form of interactive maps, both of

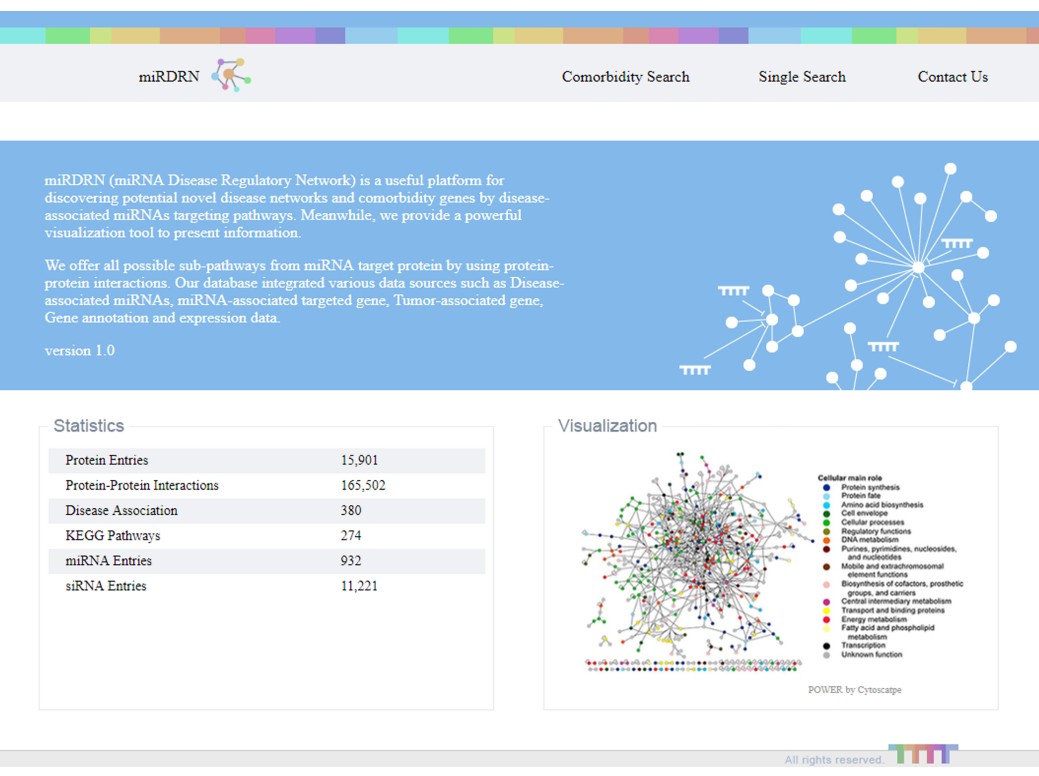

**Figure 3 Entry interface of miRDRN.** User may select "Single Search" to explore a single disease, miRNA, or siRNA, or "Comorbidity Search" to explore a disease-pair.

which may be downloaded by the user. Often a map is too large for practical visualization, and in such a case the user may use options such as setting a $p$-value cut-off, or requiring a specific gene to be present in the map, or both, to obtain a partial RRN.

The entrance interface of miRDRN (http://mirdrn.ncu.edu.tw/mirdrn/) asks the user to select "Single Search" to explore a single disease (or miRNA/siRNA) or "Comorbidity Search" to explore the comorbidity of a disease-pair (Fig. 3). The user is then asked to specify the disease or disease-pair to be explored and tissue/tumor types, and $p$-value threshold for RSP evaluation, and to click on (or not) several optional filters, respectively, on targets and on RSPs. The filter on miRNA targets allows the user to admit only targets positively validated by the seven direct experimental methods: HITS-CLIP, PAR-CLIP, IMPACT-Seq, CLASH, Luciferase Reporter Assay, 3LIFE, and Genetic Testing (*Vlachos et al., 2015*); filters on RSP allow the user to select only those RSPs with some or all of the proteins to be cancer related (Fig. 4). The user may then click on "Query" to start the computation. Tabulated results of disease-associated miRNAs and their target genes (Fig. 5), a multi-page list of all RSPs (Fig. 6) and, in the case of Comorbidity Search, a list of all comorbid genes (Fig. 7) will then automatically appear. After the first, automatic iteration, the user may reduce the size of the RSP-list by using the "Gene filter" and "Show top … sub-pathways" options (Fig. 6). The next interface (Fig. 8), in ready mode on first appearance, waits for the user to select one of three network layouts: "Tree," "Circle," or "Radial." After "Go" is clicked on, the platform displays an interactive map showing the

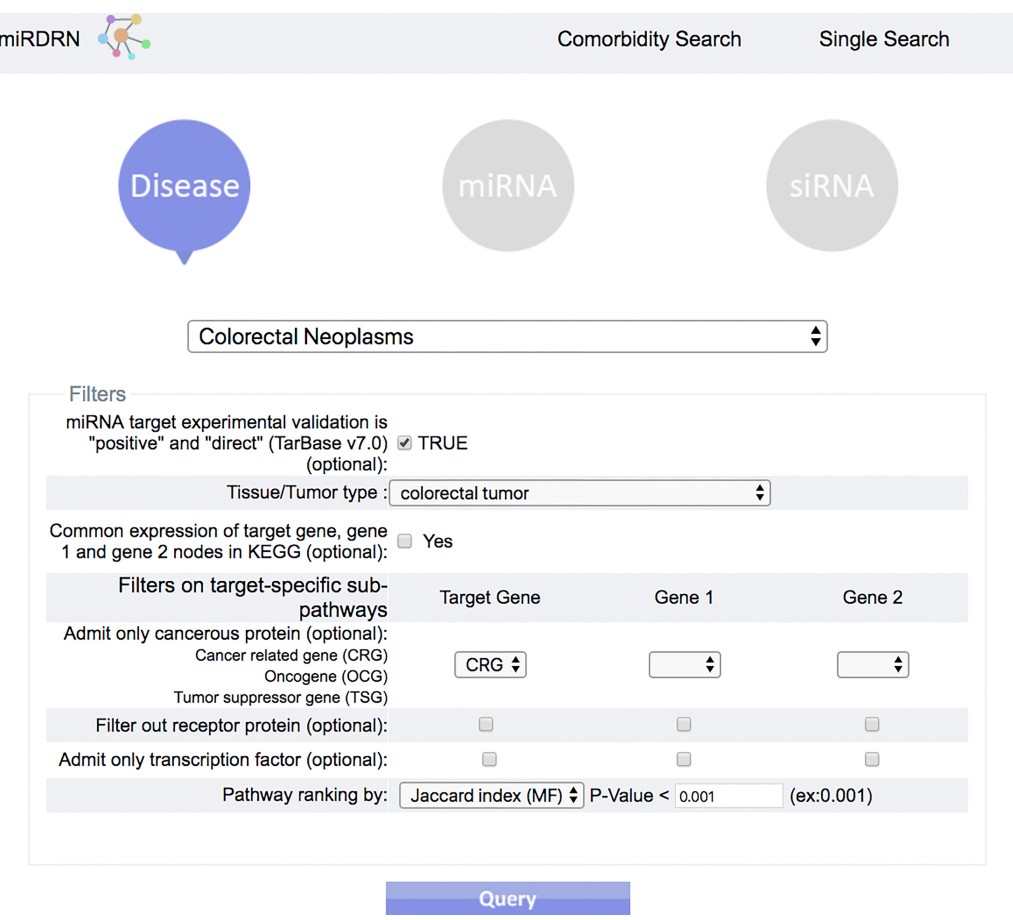

**Figure 4 Query interface of Single Search.** User is required to select a disease (or miRNA/siRNA) and other filters/options. As shown in the figure, the disease "colorectal neoplasms" with the optional tissue type "colorectal tumor" are selected. Other selections require target experimental validation to be "positive" and "direct," targets restricted to be cancer related genes, pathway ranked by Jaccard scores on molecular functions (MF), and *p*-value less than 0.001.

RRN built from RSPs selected by user-specified options (Fig. 8). When the mouse is placed on a node (a miRNA or a gene) on the map a small pop-up window opens to show the name of the node/gene and the number of other nodes it is linked to, and annotation on the node from GO, OMIM, KEGG, and GeneBank databases.

## DISCUSSION

Here, we demonstrate the utility of miRDRN by presenting three applications.

### Case 1. A single disease study of colorectal neoplasm

Here, we demonstrate a single disease application of miRDRN. On the query interface select "colorectal neoplasms" (or colorectal cancer (CRC)), tissue type "colorectal tumor," pathway ranking by "Jaccard index (MF)," and *p*-value < "0.001," and no option on target genes or RSP. The result yielded 33 miRNAs and 37 target genes (if the option "miRNA target experimental validation is positive and direct" on the query interface was selected then there would be 20 miRNAs and 23 target genes (Fig. 5)), and 45,565 RSPs

**Figure 5 Result interface on miRNAs and target genes (for colorectal neoplasms/colorectal tumor).** Search result, based on query input shown in Fig. 4, on miRNAs and literature source (blue area) and target genes (green). For each gene the gene symbol and its OMIM id are given, as well as information on whether the protein it encodes has a cancerous protein tag: CRG, cancer related gene; OCG, oncogene; TSG, tumor suppressor gene.

involving 3,079 genes (reduced to 2,111 RSPs and 1,650 genes when target is restricted to being "positive and direct" and cancer related (Fig. 6)) (Table 4).

By default, the interface "target-specific RSPs" (Fig. 6) lists all the constructed RSPs, namely all 45,565 of them in the present case and, if requested, would present a map including all the RSPs which, however, would be impractical to visualize, not to say interact with. On the same interface are two options for displaying/using a smaller RSP set: "Gene filter," where the user can restrict the set to only those RSPs containing a specified gene; and "Show top ... sub-pathways," where the user can ask for only the top-$N$ RSPs having the smallest $p$-values be listed and used for network construction. The interface "Disease specific RRN" then allows the user to choose one among the layouts "Tree," "Circle," and "Radial." Here, a tree-map, with several disconnected parts, built from the top-70 RSPs is shown (Fig. 9).

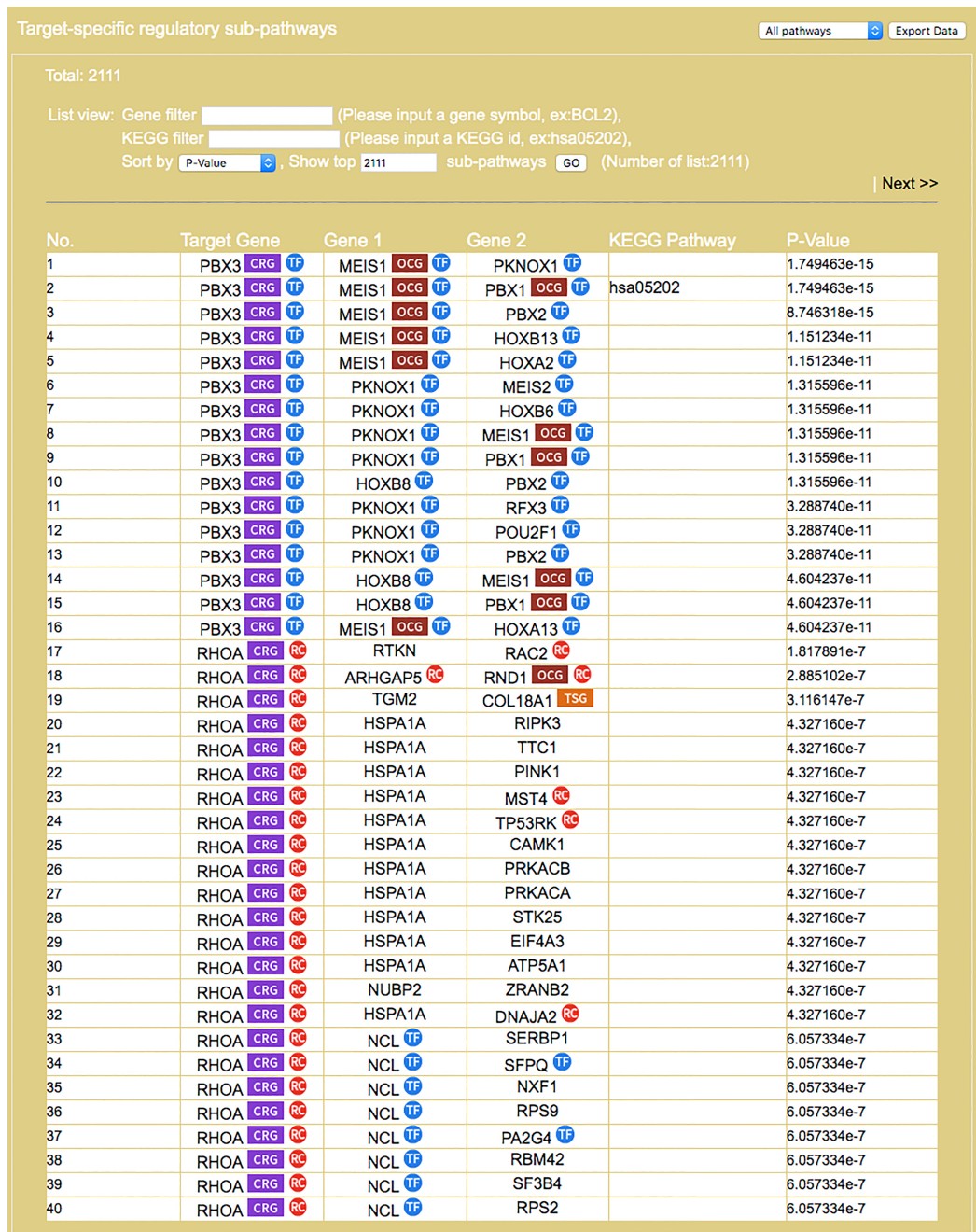

**Figure 6** **Result interface on target-specific RSPs (for colorectal neoplasms/colorectal tumor).** RSPs are listed in descending order (column 1) by *p*-value (column 6). Columns 2–4 give the symbols of genes in the sequence ($T$, $G_1$, $G_2$). Column 5 gives known pathways, such as a KEGG pathway, of which ($T$, $G_1$, $G_2$) is a part. On first appearance, all RSPs (2,111 in this example) are listed on multiple pages. Three options allow restricting the output to a smaller set: "Gene filter," where user can restrict the set to only those RSPs containing a specified gene, similarly "KEGG filter," and "Show top . . . sub-pathways," where user can ask for only the top-*N* RSPs having the smallest *p*-values be listed and used for network construction.

**Figure 7 Result on comorbidity genes in Alzheimer's disease-Type 2 diabetes comorbidity search.** Genes common to some RSPs of both diseases are listed, together with information on cancer genes status, OMIM Id, and KEGG pathway.

The largest connected sub-RRN, or "Network-1" (Fig. 10), is composed of six miRNAs targeting four genes connected to 52 other genes (Table 5). Of the 56 genes in Network-1, 22 have known CRC connections (CORECG database, http://lms.snu.edu.in/corecg) (*Agarwal et al., 2016*), and 26 others have references linking them either directly or indirectly to CRC (*Kang et al., 2016*; *Sirvent et al., 2010*; *Jeong et al., 2018*; *Xie et al., 2012*; *Wu, Wu & Jiang, 2017*; *Xiang, Wang & Xiang, 2014*; *Masuda & Yamada, 2017*; *Ali et al., 2014*; *Yun et al., 2018*; *Vázquez-Cedeira & Lazo, 2012*; *Zhang et al., 2009*; *Csukasi et al., 2018*; *Rey et al., 2016*; *Goyal et al., 2009*; *Sabir et al., 2015*; *Bjerrum et al., 2014*; *Hanna et al., 2018*; *Record et al., 2010*; *Zhou et al., 2017*; *Gong et al., 2018*; *Yuan et al., 2018*; *Kim et al., 2015*; *Li et al., 2013*; *Guo et al., 2012*; *Peng et al., 2014*; *Alonso et al., 2017*; *Qi & Ding, 2018*; *Jin et al., 2015*) (Table 6). Among these, *TNIK* (*Masuda & Yamada, 2017*) and *TNK2* (*Qi & Ding, 2018*) have been used as drug targets for CRC treatment. We consider the remaining eight genes—*PRKACA, MAP3K12, LRRK1, RIOK2, OXSR1, CDK17, EIF2AK1, TSSK4*—to be potential novel CRC-related genes. Noticeably, Network-1 has two parts, one 28 nodes (five miRNAs targeting three genes) and the other 34 nodes (one miRNA targeting one gene), connected by a single link, or PPI. The three types of

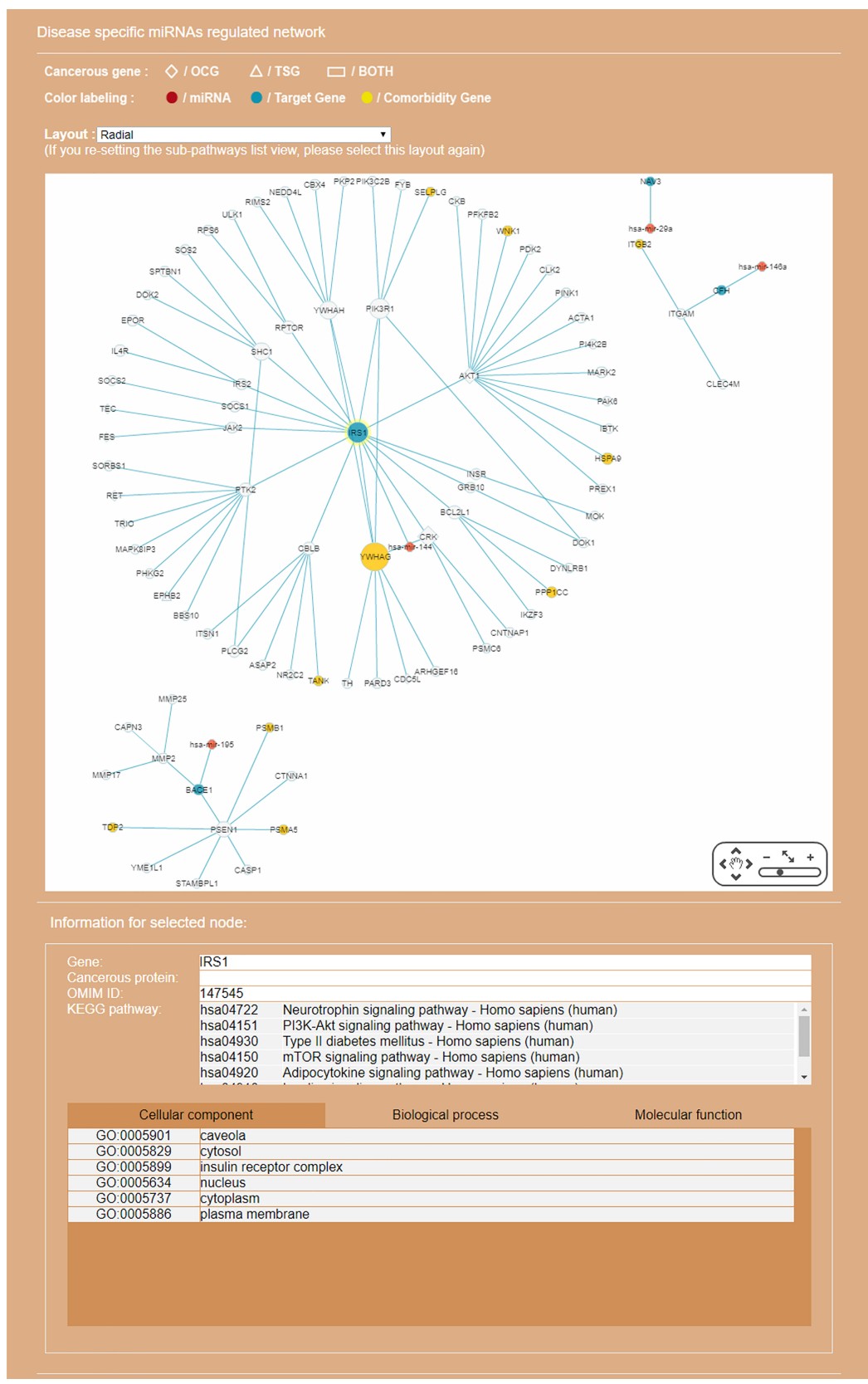

**Figure 8 Display of a sub-RRN built from a subset of RSPs determined by the user using options available in the interface shown in Fig. 6.** The option "Show top-70" RSPs (by *p*-value) was used. When the mouse is placed on a node (in this case the gene IRS1) in the displayed network, a small pop-up window opens to show the name of the node/gene and the number of other nodes it is linked to, and annotation on the node from GO, OMIM, KEGG, and GeneBank databases.

**Table 4 Result for sample Single Search: disease, colorectal neoplasm; tissue type, colorectal.**

|  | Disease |
|---|---|
| Disease name | Colorectal neoplasms |
| Tissue filter | Colorectal tumor |
| Associated miRNAs (total 33) | hsa-mir-491, hsa-mir-185, hsa-mir-20a, hsa-mir-221, hsa-mir-199a, hsa-mir-34a, hsa-mir-199b, hsa-mir-34c, hsa-mir-34b, hsa-mir-148a, hsa-mir-342, hsa-mir-21, hsa-mir-499a, hsa-let-7c, hsa-mir-148b, hsa-mir-1915, hsa-mir-17, hsa-mir-320a, hsa-mir-200c, hsa-mir-143, hsa-mir-139, hsa-mir-103a, hsa-mir-103b, hsa-mir-107, hsa-mir-497, hsa-mir-106a, hsa-mir-429, hsa-mir-7, hsa-mir-362, hsa-mir-330, hsa-mir-367, hsa-mir-339, hsa-mir-133a |
| Targeted genes (total 37) | BCL2L1, RHOA, CDC42, BNIP2, CDKN1C, AXL, MYC, BCL2, DNMT1, RHOB, FOXO4, PDCD4, MMP11, PBX3, CCKBR, CCL20, RND3, NRP1, ZEB1, CTNNB1, MACC1, IGF1R, DAPK1, KLF4, RAP1B, TGFBR2, SOX2, YY1, RBL2, E2F1, USF2, PTPN1, RYR3, PLRG1, RFFL, DNMT3A, KRAS |
| Regulatory sub-pathways | 45,565 |
| Distinct genes | 3,079 |

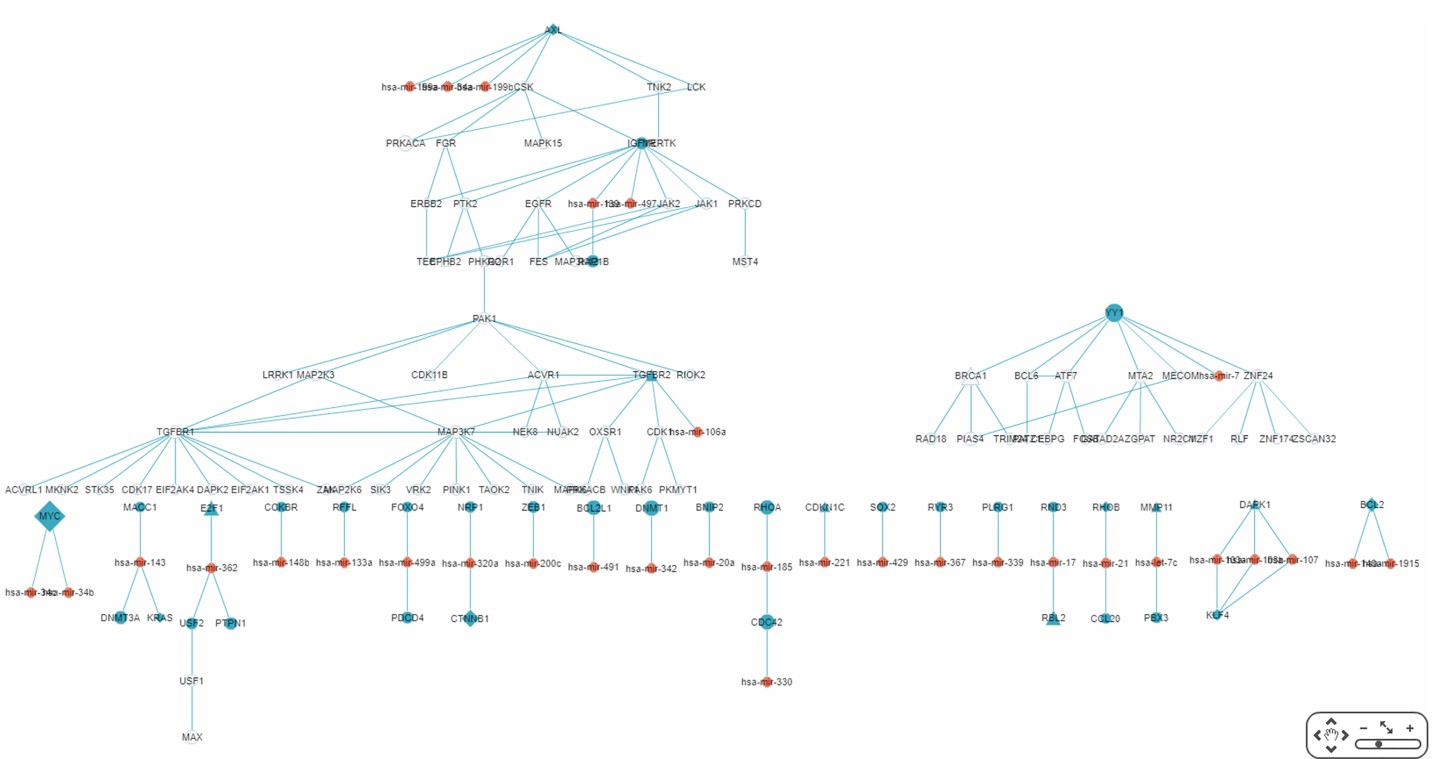

**Figure 9 A partial miRNA regulatory network (RRN) for colorectal neoplasm.** The RRN is constructed from the top 70 RSPs by *p*-value for colorectal neoplasm, tissue type, colorectal tumor. A link indicates a miRNA-target relation or a PPI; red circle, miRNA; blue circle, miRNA target gene; yellow circle, non-target gene; diamond, oncogene; triangle, tumor suppressor gene.

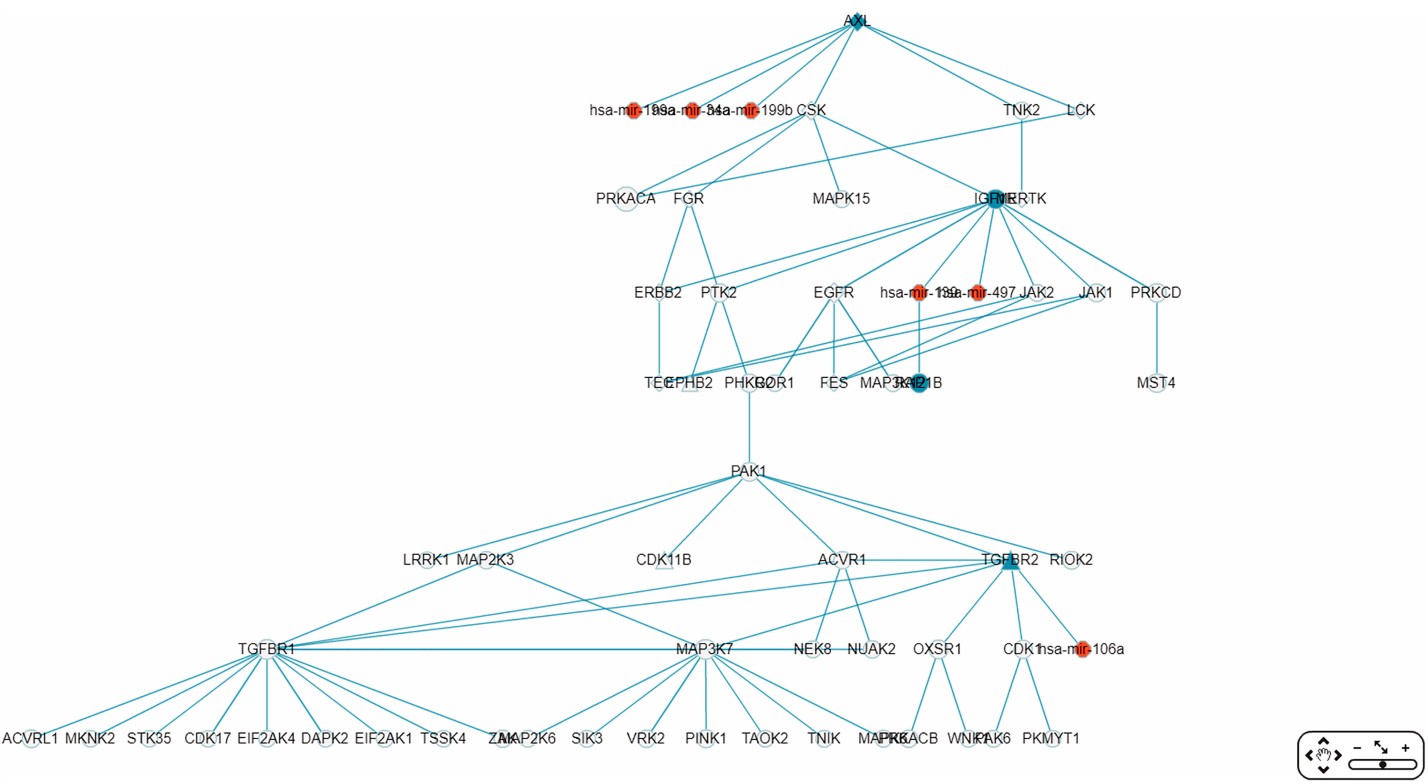

**Figure 10 The sub-RRN network-1.** This largest connected sub-RRN for colorectal neoplasm (constructed from the top 70 RSPs by *p*-value), containing six miRNAs targeting four genes connected to 52 other genes, is itself composed of two parts, one 28 nodes (five miRNAs targeting three genes) and the other 34 nodes (one miRNA targeting one gene), connected by a single link.

**Table 5 Statistics and gene information in the network-1, the largest connected sub-network of the CRC-specific miRNA regulatory network.**

| | | Number | Item set |
|---|---|---|---|
| Network-1 | miRNAs | 6 | hsa-mir-199a, hsa-mir-34a, hsa-mir-199b, hsa-mir-139, hsa-mir-497, hsa-mir-106a |
| | Target genes | 4 | AXL, IGF1R, RAP1B, TGFBR2 |
| | Gene set (including target genes) | 56 | AXL, CSK, TNK2, LCK, PRKACA, FGR, MAPK15, IGF1R, MERTK, ERBB2, PTK2, EGFR, JAK2, JAK1, PRKCD, TEC, EPHB2, PHKG2, ROR1, FES, MAP3K12, RAP1B, MST4, PAK1, LRRK1, MAP2K3, CDK11B, ACVR1, TGFBR2, RIOK2, TGFBR1, MAP3K7, NEK8, NUAK2, OXSR1, CDK1, ACVRL1, MKNK2, STK35, CDK17, EIF2AK4, DAPK2, EIF2AK1, TSSK4, ZAK, MAP2K6, SIK3, VRK2, PINK1, TAOK2, TNIK, MAPK6, PRKACB, WNK1, PAK6, PKMYT1 |

genes, known CRC-related, reference-supported, and potential CRC-related, are more or less proportionately distributed in these two parts.

The "Gene filter" option (Fig. 6) allows the user to focus on a specific gene in RRN construction. As example, *TNK2*, a key drug target for the treatment of metastatic CRC (*Qi & Ding, 2018*), was selected as the filter, together with the "Show top 70 RSPs" option. The result was a nine-node sub-RRN: the target gene *AXL* regulated by three miRNAs—hsa-mir-199b, hsa-mir-34a, hsa-mir-199a—and linked (by PPI) to *TNK2*, itself linked to four other genes *AXL*(OCG), *MAGI3, HSP90AB2P, MERTK*(OCG), *KAT8* (Fig. 11).

**Table 6 Known, literature supported, and potential novel CRC-related genes.**

| | | Number | Item set |
|---|---|---|---|
| Gene set (Network-1) | Known CRC genes | 22 | AXL, LCK, FGR, IGF1R, MERTK, ERBB2, PTK2, EGFR, JAK2, JAK1, EPHB2, FES, PAK1, MAP2K3, ACVR1, TGFBR2, TGFBR1, CDK1, EIF2AK4, DAPK2, MAP2K6, PAK6 |
| | Reference supported (*Kang et al., 2016*; *Sirvent et al., 2010*; *Jeong et al., 2018*; *Xie et al., 2012*; *Wu, Wu & Jiang, 2017*; *Xiang, Wang & Xiang, 2014*; *Masuda & Yamada, 2017*; *Ali et al., 2014*; *Yun et al., 2018*; *Vázquez-Cedeira & Lazo, 2012*; *Zhang et al., 2009*; *Csukasi et al., 2018*; *Rey et al., 2016*; *Goyal et al., 2009*; *Sabir et al., 2015*; *Bjerrum et al., 2014*; *Hanna et al., 2018*; *Record et al., 2010*; *Zhou et al., 2017*; *Gong et al., 2018*; *Yuan et al., 2018*; *Kim et al., 2015*; *Li et al., 2013*; *Guo et al., 2012*; *Peng et al., 2014*; *Alonso et al., 2017*; *Qi & Ding, 2018*; *Jin et al., 2015*) | 26 | CSK, TNK2*, MAPK15, PRKCD, TEC, PHKG2, ROR1, RAP1B, MST4, CDK11B, MAP3K7, NEK8, NUAK2, ACVRL1, MKNK2, STK35, ZAK, SIK3, VRK2, PINK1, TAOK2, TNIK*, MAPK6, PRKACB, PKMYT1, WNK1 |
| | Potential novel cancer-related gene | 8 | PRKACA, MAP3K12, LRRK1, RIOK2, OXSR1, CDK17, EIF2AK1, TSSK4 |

**Note:**
* Known target genes used in treatment of CRC.

## Case 2. A comorbidity study of the disease-pair Alzheimer's disease-Type 2 diabetes

Recent studies suggest a possible AD-T2D comorbidity. Known pathophysiological factors shared by AD and T2D include insulin, cholesterol, β-amyloid aggregation and tau (*Akter et al., 2011*). High cholesterol level impacts β-amyloid formation in the brain (*Reed et al., 2014*); abnormal insulin function, a key factor of T2D and related disorders (*Basu et al., 2005*), increases the risk for AD (*Ronnemaa et al., 2008*); cardiovascular risk factors such as high cholesterol and hypertension are common to T2D and AD (*Kivipelto et al., 2005*). Evidences connecting AD to impaired function of insulin/IGF suggested AD might be viewed as a new type, "type 3," of diabetes (*Lester-Coll et al., 2006*). However, another study claims T2D to be associated with cerebrovascular but not Alzheimer neuropathology (*Abner et al., 2016*). Here, we demonstrate a two-disease application of miRDRN. After logging onto miRDRN's entry interface (Fig. 3), click on "Comorbidity Search" to see a new interface urging the user to select two diseases; for "Disease 1," "Alzheimer disease" (AD) and tissue type "brain" were selected and for "Disease 2," "Type 2" (which stands for T2D) and tissue type "pancreas." Pathway ranking by "Jaccard index (MF)," and *p*-value < "0.005" for both diseases were selected. Both AD and T2D are complex diseases and share aging for a risk factor; accumulated evidence indicates a connection between these two diseases at the molecular level (*Ahmed et al., 2014*). For this case miRDRN yielded, for AD (T2D), three (one) associated-miRNAs, three (one) targeted genes, 644 (3,908) RSPs, involving 633 (2,187) genes (Table 7). Because AD and T2D did not have any common associated-miRNA target gene, they had distinct sets of RSPs. The 500 genes common to the two sets of RSPs (25 of which are

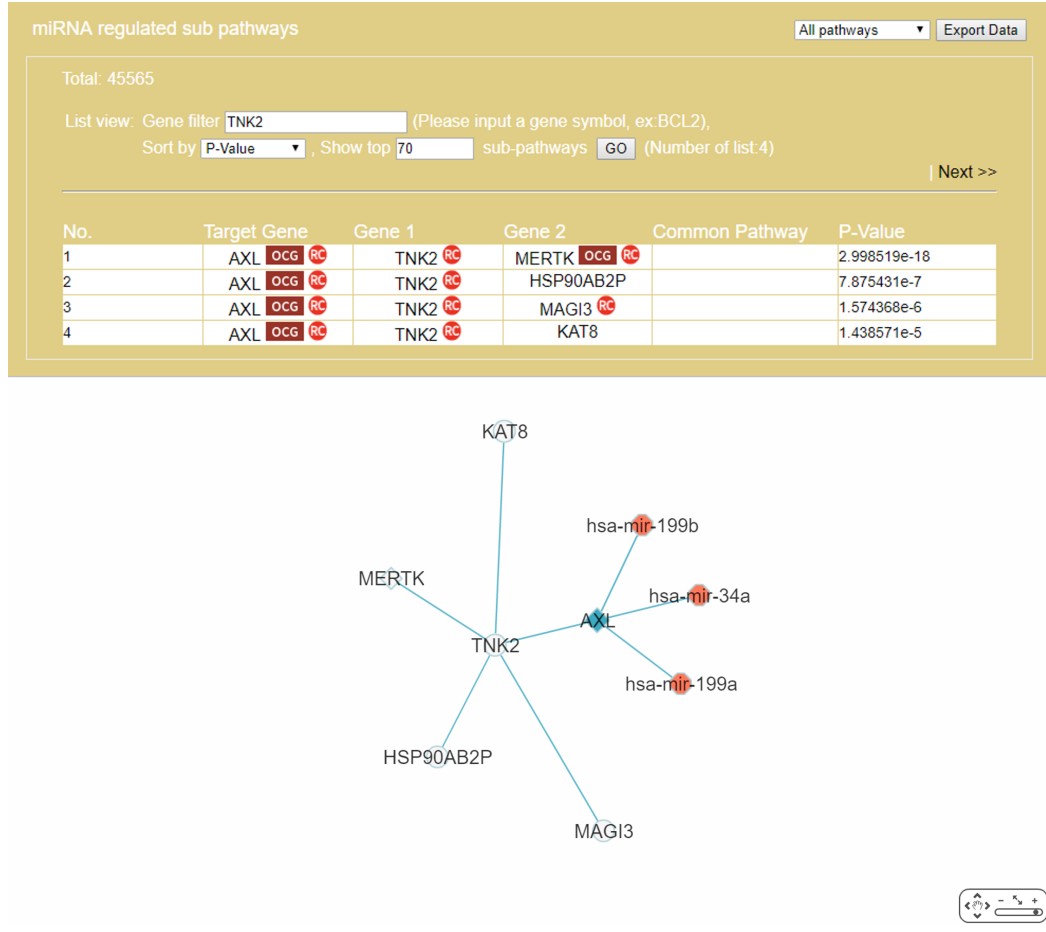

**Figure 11 A sub-RRN of CRC obtained by using *TNK2* as a gene filter.** The RRN contains the target gene *AXL* regulated by three miRNAs, hsa-mir-199b, hsa-mir-34a, hsa-mir-199a, and linked by PPI to *TNK2*, itself linked by PPI to four other genes *AXL*(OCG), *MAGI3*, *HSP90AB2P*, *MERTK*(OCG), *KAT8*.

shown in Fig. 7) are significantly enriched in three KEGG terms: hsa03040:Spliceosome ($p$-value = 0.00549), hsa03018:RNA degradation ($p$-value = 0.00802), and hsa03022: Basal transcription factors ($p$-value = 0.00415). Abnormality of spliceosome has been reported in both AD (*Love, Hayden & Rohn, 2015*) and T2D (*Dlamini, Mokoena & Hull, 2017*) patients. Among the comorbid genes, 8—*ALOX5, APP, BIN1, CHGB, VWF, NEFL, LETMD1, CELF1*- were identified as known AD target genes (*Bertram et al., 2007*; *Sun et al., 2012*; *Bai et al., 2016*) and 14—*TCF7L2, APOA1, VWF, CDKN2B, CAT, ITGB2, ISL1, POLD3, APP, NFKBIB, GNA12, DEDD, LDLR, PRKAB1*- as known T2D target genes (*Dai et al., 2013*), *APP* and *VWF* are known targets of both diseases (Table 8). With the exception of three—*LEMD1, POLD3, GNA12*, the comorbidity of all the others have literature support (Table 8).

## Case 3. A sub-RRN centered on the AD-associated gene *BACE1*

In recent years a number of anti-AD drugs designed on the basis of the amyloid-beta (Aβ) hypothesis of AD, which holds that Aβ aggregate in the brain is the main causative factor

**Table 7  Results for the AD-T2D comorbidity study.**

|  | Disease 1 | Disease 2 | Comorbidity |
|---|---|---|---|
| Disease name | AD | T2D | AD/T2D |
| Tissue filter | Brain | Pancreas | Brain/pancreas |
| Associated-miRNA | hsa-mir-29a, hsa-mir-195, hsa-mir-146a | hsa-mir-144 | hsa-mir-29a, hsa-mir-195, hsa-mir-146a, hsa-mir-144 |
| Targeted gene | NAV3, BACE1, CFH | IRS1 | NAV3, BACE1, CFH, IRS1 |
| Regulatory sub-pathways | 644 | 3,908 | 4,552 |
| Total no. of genes | 633 | 2,187 | 2,320 |
| No. of common genes | – | – | 500 |

**Table 8  Known, literature supported, and potential novel AD-T2D comorbid genes.**

| | | No. of targets in comorbidity gene set (500) | Comorbid genes (references) |
|---|---|---|---|
| Known data | Known AD target (210) | 8 | ALOX5 (*Postula et al., 2016*; *Nejatian et al., 2015*; *Heemskerk et al., 2015*), APP*, BIN1 (*Greenbaum et al., 2016*), CHGB (*Horn et al., 2016*), VWF*, NEFL (*Wu et al., 1996*; *Celikbilek et al., 2014*), LETMD1#, CELF1 (*Verma et al., 2013*; *Belanger et al., 2018*) |
| | Known T2D target (497) | 14 | TCF7L2 (*Blom et al., 2010*; *Včelák et al., 2012*; *Arefin et al., 2012*; *Riise et al., 2015*), APOA1 (*Vollbach et al., 2005*; *Raygani et al., 2006*; *Lin, Cao & Gao, 2015*), VWF*, CDKN2B (*Pallàs et al., 2005*), CAT (*Gsell et al., 1995*; *Lovell et al., 1995*; *Kim et al., 2008*), ITGB2 (*Mizwicki et al., 2013*; *Cifuentes & Murillo-Rojas, 2014*), ISL1 (*Yang et al., 2016*), POLD3*, APP*, NFKBIB (*Li et al., 2015*), GNA12#, DEDD (*Engidawork et al., 2001*; *Reed 2000*), LDLR (*Gopalraj et al., 2005*; *Bowen, Isley & Atkinson, 2000*), PRKAB1 (*Vingtdeux et al., 2011*; *Greco et al., 2009*; *Salminen et al., 2011*) |

Notes:
The 210 known AD target genes were built by integrating gene lists from AlzGene (*Bertram et al., 2007*), AlzBIG (*Sun et al., 2012*) and AlzBase (*Bai et al., 2016*); the 497 known T2D targets were from T-HOD (*Dai et al., 2013*).
* Known AD and T2D target.
# No literature support.

of AD, failed late-phase trials. These include the γ-secretase inhibitor Semagacestat (*Fleisher et al., 2008*) and two *BACE1* inhibitors, Verubecestat (*Kennedy et al., 2016*) and Atabecestat (*Timmers et al., 2016*). In all three cases treatment groups scored worse than the control group on the Alzheimer's disease cooperative study activities of daily living inventory (ADCS-ADL) functional measure and reported more anxiety, depression, and sleep problems than controls. In a "Single Search" application on AD (tissue, brain; $p$-value threshold, 0.005), we had miRDRN construct a partial RRN (Gene filter, *BACE1*; Show top 70 sub-pathways; Network layout, Radial) centered on *BACE1*, which is a regulatory target of hsa-mir-195. The result shows the genes *PSEN1*, *NCSTN*, *RANBP9*, *PLSCR1*, *MMP2*, and *FURIN* to be immediately downstream to *BACE1* in the RRN (Fig. 12). *PSEN1* and *NCSTN* encode proteins that are, respectively, catalytic and essential subunits of the γ-secretase complex; suppression of these genes are presumably the purpose of *BACE1* inhibition. On the other hand, *RANBP9* encodes a protein that

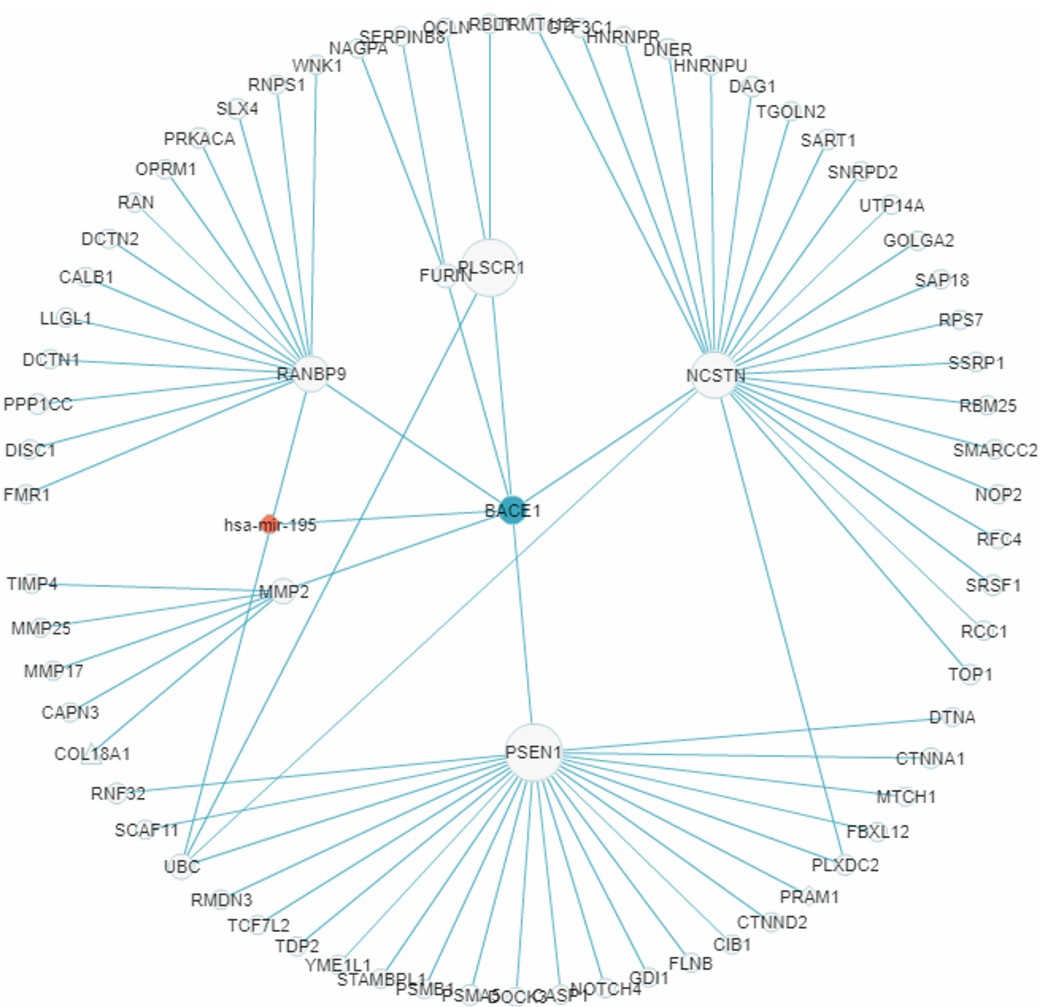

**Figure 12 A miRNA regulatory sub-network centered on the AD-associated gene *BACE1*.** The genes *PSEN1*, *NCSTN*, *RANBP9*, *PLSCR1*, *MMP2*, and *FURIN* are shown to be immediately downstream to, that is, have level 1 PPI with, *BACE1*.

facilitates the progression of mitosis in developing neuroepithelial cells (*Chang et al., 2010*); *PLSCR1* encodes a protein that acts in the control of intracellular calcium homeostasis and has a central role in signal transduction (*Tufail et al., 2017*); *MMP2* encodes a protein that promotes neural progenitor cell migration (*Rojiani et al., 2010*). Suppression of these genes (by *BACE1* inhibition) may therefore adversely affect signal transduction and the nerve system, and could be part of the reason why Semagacestat, Verubecestat, and Atabecestat worsened the ADCS-ADL functional measure of treatment groups.

## CONCLUSION

This work describes miRDRN (http://mirdrn.ncu.edu.tw/mirdrn/), composed of a new database on target-specific RSPs and a web service platform that allows the user to use the stored RSPs to construct disease and tissue-specific RRNs, which may aid the user to explore disease related molecular and pathway associations, or find new ones.

As demonstration, miRDRN was applied to study the single disease CRC, where 34 potential target genes were identified, 26 of which have literature support; to study the comorbidity of the disease-pair AD-T2D, where 20 potential novel AD-T2D comorbid genes were identified, 17 of which have literature support; and to construct a partial miRNA regulatory sub-network centered on the AD-associated gene *BACE1*, which in turn suggests a possible explanation why, in late-phase trials that ended in failure, several γ/β-secretase inhibiting anti-AD drugs worsened the functional measure of treatment groups. We believe that findings from miRDRN, even exploratory in nature, may potentially lead to the identification of new drug targets and new understanding in modes of drug action.

## ABBREVIATIONS

| | |
|---|---|
| **AD** | Alzheimer's disease |
| **BioGrid** | biological general repository for interaction datasets |
| **CRG** | cancer related gene |
| **GO** | gene ontology database |
| **HMDD** | human microRNA and disease associations database |
| **KEGG** | Kyoto encyclopedia of genes and genomes |
| **miRDRN** | miRNA disease regulatory network database and web service platform |
| **MRSP** | miRNA-specific regulatory sub-pathway |
| **OCG** | oncogene |
| **PPI** | protein–protein interaction |
| **PPIN** | PPI network |
| **RSP** | target-specific regulatory sub-pathway |
| **RRN** | disease-specific miRNA regulatory network |
| **T2D** | Type 2 diabetes |
| **TarBase** | database on miRNA:mRNA interactions |
| **TSG** | tumor suppressor gene. |

### Funding

This work was supported by the Ministry of Science and Technology, Republic of China (No. MOST-105-2314-B-033-001). The funders had no role in study design, data collection and analysis, decision to publish, or preparation of the manuscript.

### Grant Disclosures

The following grant information was disclosed by the authors:
Ministry of Science and Technology, Republic of China: MOST-105-2314-B-033-001.

### Competing Interests

The authors declare that they have no competing interests.

## Author Contributions

- Hsueh-Chuan Liu conceived and designed the experiments, performed the experiments, analyzed the data, contributed reagents/materials/analysis tools, prepared figures and/or tables, authored or reviewed drafts of the paper, approved the final draft.
- Yi-Shian Peng authored or reviewed drafts of the paper, approved the final draft, provided material and did literature search for discussion in the manuscript related to AD, T2D, and anti-AD drugs.
- Hoong-Chien Lee conceived and designed the experiments, analyzed the data, authored or reviewed drafts of the paper, approved the final draft.

## Data Availability

The source code and data of miRDRN is available at and freely downloadable from the public repository GitHub: https://github.com/o2snow/2019MIRDRN.

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
