# Peer review of "miRDRN—miRNA disease regulatory network: a tool for exploring disease and tissue-specific microRNA regulatory networks"

_PeerJ, doi:10.7717/peerj.7309_

## Round 0.1 · original submission · Minor Revisions

Although the requested changes are minor, they are essential to improve the manuscript.

The major suggestion is to compare to similar databases or those with similar scope. The other important change suggested by the reviewers is to improve the database website, especially by revising the disease list + to add an exclusion filter.

Other useful suggestions, made by Reviewer 2, should also improve the methodology.

Finally, while the manuscript is readable and clear, some polishing and final language editing is needed— (I'm sure the authors can do that after implementing the suggested changes).

[]

·

Basic reporting

No Comment

Experimental design

No comment

Validity of the findings

Regarding the use of databases for miRNA targets, the authors relied on the HMMD database that provides data about experimentally validated miRNA targets. The web-based tool described by the authors can include a filter that allow users to exclude miRNA targets based on the experimental method used to validate those miRNA targets. (For example excluding targets validated by methods that used mRNA expression analysis by QPCR and include only targets validated on the protein level, or based on the knockout/overexpression methods for functional analysis of those miRNAs…etc). This is due to the fact that validating miRNA targets biologically can be challenging.

·

Basic reporting

The manuscript by Liu et al. describes a database, miRDRN, that compiles associations between miRNA, target genes/proteins, and diseases. The database considers multiple levels of data and computes correlation between miRNA, genes and diseases by using principles of the set theory. The data used to build miRDRN was obtained from other databases that mostly rely on experimentally-validated sources, however, the authors do not provide any comparison to previously published alike databases or even to the ones from which they obtained the data for building miRDRN (e.g. HMDD, TarBase).

Experimental design

-- line 111: how did the authors integrate the different data types ? or did they mean 'compiled' ?
-- the definition of Jaccard score for detecting the overlap between two sets of genes as explained by the authors is very informative and very well-suited to GO. However, for pathways and since the authors account for primary and secondary interactions, it will be worth weighing the genes/proteins in a pathway depending on how near/far they are from the target genes (i.e. order of steps from/to target in a pathway)
-- line 161: step 3 mentions that targets of a specific miRNA are collected from two sources (HMDD and TarBase). Does the authors take the overlapping targets from the two sources (intersection) or union ?
-- It is not clear to me how the authors integrated the miRNA-disease association data. They have elaborated on how miRNA-target associations were computed but not for the disease association. Also, it is not clear how they define co-morbidity from a computational point of view.

Validity of the findings

-- lines 292-293: Are the 26 targets that have literature support well-validated targets ? if yes, I would suggest that the authors curate a list of positive and negative association 'gold standard' targets and then use this set to quantitatively evaluate the performance of their tool.
-- The authors should provide a comparison to existing similar tools and highlight the advantages of using their tool in particular. I am not up-to-date on microRNA databases but a quick search shows there are several against which the authors should mention (e.g. HMDD, miRwayDB). Also several of these databases have experimentally validated targets, so I strongly suggest that the authors provide quantitative performance evaluation measures (AUROC, F1 score, FPR ... etc)
-- the authors should clearly hint that the findings form their tool are exploratory in nature with the goal of prioritizing miRNA-target-disease associations.

Additional comments

-- the discussion on potential failure of the anti-AD drug (case #3) is very interesting and demonstrates the utility of the database in full. I would strongly suggest that the authors follow the same for the other two case studies (cases #1, #2). The authors could also hint to potential applications of building similar databases in the introduction and highlight that their databases is very comprehensive in guiding associations across several types of data that could potentially lead to identifying new drug targets, understanding drugs mode of action ...etc.

minor comments:
-- the names of the diseases in the 'disease' drop box need to be revised, for instance 'acute', 'chronic' and 'hepatocellular' are not proper diseases names.
-- what is the default p value cutoff ?
-- line 132: the intersection and union symbols are reversed.

---

## Round 0.2 · accepted · Accept

Thank you for taking all reviews seriously, especially for addressing the comparison with other databases, and for even adding more analysis.

---

## Author Rebuttal · Round 0.2

# miRDRN –miRNA Disease Regulatory Network: A tool for exploring disease and tissue-specific microRNA regulatory networks" (#2019:04:36918:0:2:REVIEW)

Hsueh-Chuan Liu, et al.

Response to Editor's and Reviewers' comment/recommendation

## Academic Editor

1. *The major suggestion is to compare to similar databases or those with similar scope. The other important change suggested by the reviewers is to improve the database website, especially by revising the disease list + to add an exclusion filter.*

   Authors' response. On comparison with other related database please see response to comment #4 from Reviewer 2, on addition of exclusion filter please see response to comment #3 from Reviewer 1.

2. *while the manuscript is readable and clear, some polishing and final language editing is needed— (I'm sure the authors can do that after implementing the suggested changes).*

   Authors' response. Our apologies for sloppy language at places in the original manuscript. Parts in revised manuscript that have been significantly rewritten/polished for clarity are shown as highlights in the mark-up copy, and include the Abstract, sections on data integration, comparison with other related databases, preamble in AD-T2D comorbidity, and conclusion.

## Reviewer 1

3. *The web-based tool described by the authors can include a filter that allow users to exclude miRNA targets based on the experimental method used to validate those miRNA targets. (For example excluding targets validated by methods that used mRNA expression analysis by QPCR and include only targets validated on the protein level, or based on the knockout/overexpression methods for functional analysis of those miRNAs…etc). This is due to the fact that validating miRNA targets biologically can be challenging.*

   Authors' response. We thank the Reviewer for this suggestion. On the interface for disease(s) selection (Figure 4), an option has been added where user can opt for including a target only if "experimental validation is "positive" and "direct" according to TarBase v7.0".  In the manuscript, text relevant to this information occurs (in the mark-up copy) at lines 130-131 and 242-244.

## Reviewer 2

### Basic reporting
4. *the authors do not provide any comparison to previously published alike databases or even to the ones from which they obtained the data for building miRDRN (e.g. HMDD, TarBase).*

Authors' response. In the revised manuscript, miRDRN is compared with five other miRNA-related databases: HMDD, TarBase, PhenomiR, miRPathDB, and miRwayDB in a new section (mark-up copy lines 215-224) and (a new) Table 3. Content provided by miRDRN but not by any of the five databases include: a database on 6,973,875 $p$-valued target-specific regulatory sub-pathways and a web service platform that facilitates the construction and visualization of disease-specific regulatory networks and downloading of related output.

## Experimental design

5. *line 111: how did the authors integrate the different data types? or did they mean 'compiled'?*

   Authors' response. Data integration is now described in detail the section Data integration (mark-up copy lines 124-142).

6. *The definition of Jaccard score for detecting the overlap between two sets of genes as explained by the authors is very informative and very well-suited to GO. However, for pathways and since the authors account for primary and secondary interactions, it will be worth weighing the genes/proteins in a pathway depending on how near/far they are from the target genes (i.e. order of steps from/to target in a pathway)*

   Authors' response. We thank the Reviewer for this suggestion, which should enhance the quality of the Jaccard score. However, because the scope of the computation increases exponentially with the number of steps, we believe the 1.0 version of miRDRN does not warrant such a major undertaking. In the future, based on user demand, we may implement this improvement in an update of miRDRN.

7. *line 161: step 3 mentions that targets of a specific miRNA are collected from two sources (HMDD and TarBase). Does the authors take the overlapping targets from the two sources (intersection) or union?*

   Authors' response. The assembly of the 6,973,875 $p$-valued target-specific regulatory sub-pathways (RSPs), now stored in miRDRN, used the union set of targets from HMDD and TarBase (mark-up copy lines 188-192).

8. *It is not clear to me how the authors integrated the miRNA-disease association data. They have elaborated on how miRNA-target associations were computed but not for the disease association.*

   Authors' response. The user-initiated construction of disease-specific regulatory networks uses only miRNA-target associations from HMDD (mark-up copy lines 127-128, 196-197), and non-disease-specific networks, from TarBase (lines 128-129). Both use the set of RSPs stored in miRDRN.

## Validity of findings

9. *It is not clear how they define co-morbidity from a computational point of view.*

Authors' response. miRDRN provides information on co-morbidity by identifying genes common to the disease-specific miRNA-protein regulatory networks of two diseases. These genes may be downloaded by the reader (the Export Data button on the "Comorbidity Gene" interface, Figure 7). These genes can be used, for instance, to identify biological functions in which they are significantly enriched. See mark-up copy lines 328-333.

10. *lines 292-293: Are the 26 targets that have literature support well-validated targets?*

Authors' response. We assume the Reviewer is referring to the 26 genes that the manuscript says "have references linking them either directly or indirectly to CRC [28-55] (Table 6, previously Table 5)." (mark-up copy lines 287-288). We are not saying whether or not the literature provides support that the genes are well-validated targets.

11. *The authors should provide a comparison to existing similar tools and highlight the advantages of using their tool in particular. I am not up-to-date on microRNA databases but a quick search shows there are several against which the authors should mention (e.g. HMDD, miRwayDB). Also several of these databases have experimentally validated targets, so I strongly suggest that the authors provide quantitative performance evaluation measures (AUROC, F1 score, FPR ... etc)*

Authors' response. For comparison see response to comment #4; for target validation see response to comment #3.

12. *The authors should clearly hint that the findings form their tool are exploratory in nature with the goal of prioritizing miRNA-target-disease associations.*

Authors' response. We have added such language in the revised manuscript; see mark-up copy lines 119-121, 371-372, 380-382.

Comments for the Author
13. *The discussion on potential failure of the anti-AD drug (case #3) is very interesting and demonstrates the utility of the database in full. I would strongly suggest that the authors follow the same for the other two case studies (cases #1, #2). The authors could also hint to potential applications of building similar databases in the introduction and highlight that their databases is very comprehensive in guiding associations across several types of data that could potentially lead to identifying new drug targets, understanding drugs mode of action ...etc.*

Authors' response. For case #2, we have added results on the identification of KEGG terms enriched in the set of 500 AD-T2D "comorbid genes; see mark-up copy lines 328-333. See also response to previous comment.

Minor comments
14. *The names of the diseases in the 'disease' drop box need to be revised, for instance 'acute', 'chronic' and 'hepatocellular' are not proper diseases names.*

Authors' response. These terms were directly taken from HMDD 2.0.